# Bolus Effect Caused by Use of Thermoplastic Masks in Head and Neck Radiotherapy Treatments

**DOI:** 10.3390/ijms25169133

**Published:** 2024-08-22

**Authors:** Diego A. Barajas-Lopez, Cristian C. Castellanos-Jerez, José A. Diaz-Merchan, S. A. Martinez-Ovalle

**Affiliations:** 1Applied Nuclear Physics and Simulation Research Group, Pedagogical and Technological University of Colombia, Tunja 150003, Colombia; diego.barajas@uptc.edu.co (D.A.B.-L.);; 2Radiotherapy Department, Clínica de las Américas AUNA, Medellin 050032, Colombia; 3Medical Physics Department, Clínica Cancerológica de Boyacá, Tunja 150003, Colombia

**Keywords:** thermoplastic mask, bolus effect, radiochromic film, MCNPX

## Abstract

This paper examines the dosimetric uncertainty arising from the use of thermoplastic masks in the treatment of head and neck cancer through radiotherapy. This study was conducted through Monte Carlo simulations using the Monte Carlo N-Particle eXtended (MCNPX code), and the theoretical results are compared with radiochromic films. Using material characterization techniques, the compounds of the thermoplastic mask were identified, confirming that most of the material corresponds to the polymer C_10_H_16_O_4_. The theoretical results show increases ranging from 42% to 57.4% in the surface absorbed dose for 6 and 15 MV photon beams, respectively, compared to the absorbed dose without the mask. The experimental data corroborate these findings, showing dose increases ranging from 18.4% to 52.1% compared to the expected surface absorbed dose without the mask. These results highlight the need to consider the bolus effect induced by thermoplastic masks during the precise and safe planning and application of radiotherapy treatment in order to ensure its therapeutic efficacy and minimize the associated risks to patients.

## 1. Introduction

Head and neck cancer has a significant incidence, representing 9.19% of reported cases in 2020 according to the Global Cancer Observatory [1]. This high incidence underscores the importance of optimizing treatment techniques such as radiotherapy, chemotherapy, and surgical interventions [2].

In external radiotherapy, immobilizers, such as thermoplastic masks, are crucial for maintaining treatment precision and avoiding damage to organs at risk (OARs) due to involuntary patient movements [3].

Thermoplastic masks are polymers that become elastic when reaching temperatures between 65 °C and 75 °C, allowing precise conformation to the patient’s anatomy for use during radiotherapy sessions [3]. These masks are mainly made of polycaprolactone (PCL) and may contain additional elements such as sulfur to reduce the vitrification temperature and make the material more manageable at lower temperatures [4]. The density of these masks can vary between 0.9 and 2 g·m^−3^ [5,6].

Studies have confirmed how the use of thermoplastic masks can increase the radiation dose on the skin surface, raising the risk of burns and associated morbidity. Lee et al. [7] used thermoluminescent dosimeters and found that the masks can significantly increase the surface dose when the treatment is performed with 6 MV photons, with an increase of 8.7% to 31.5%. Halm et al. [8] concluded that cobalt therapy generates significant increases and that 6 MeV photons increase the surface dose by 13.5% at 0.5 mm depth and 11.5% in the first millimeter. Hadley et al. [9] observed that for 15 MV photons, the surface dose can increase by between 6% and 22% and up to 36% with 6 MV photons, depending on factors such as beam orientation, energy, and mask thickness. Soleymanifard et al. [10] found significant increases in surface dose with 6 MV photons, between 20% and 32%. These authors also state that for photons of 15 MV, the enhancement can be ignored, a debatable conclusion in the results presented by other authors.

The increase in surface absorbed dose due to materials overlapping the skin during the radiotherapy session is known as the bolus effect. This effect is beneficial in skin cancer treatments as it superficializes the dose, increasing its effectiveness and reducing the exposure of organs at risk [11]. However, in the treatment of head and neck cancer, the bolus effect is usually not estimated, which can lead to undesired side effects.

Given the potential impact of the bolus effect on dosimetry, this paper aims to determine the dosimetric uncertainty caused by the use of thermoplastic masks in head and neck cancer treatments, thereby contributing to the optimization and safety of radiotherapy procedures.

## 2. Results

Figure 1 shows the experimental setup of the phantom with and without mask. These two configurations are reproduced by Monte Carlo simulations.

Figure 2 shows the Percentage Depth Dose (PDD) results in the first mm of depth for energies of 6 and 15 MV. The green curve corresponds to the PDD without a mask, the blue curve corresponds to condition #1, and the orange curve shows condition #2, as described in Table 1.

From the analysis of Figure 2, the following is evident: Condition #1 does not show significant differences compared to the calculation without a mask. On the other hand, the differences between the no-mask condition and condition #2 are notable for 6 and 15 MV energies.

Table 1 presents the most representative numerical values from the analysis of Figure 2, allowing differences to be established between the three conditions studied in the simulations. The important values to analyze correspond to the position of the maximum absorbed dose point and the PDD value on the surface, considering 1.5 mm depth as a reference, which is the average skin thickness [12].

Analyzing Table 1 for the three cases studied, it was found that the minimum associated uncertainty, both for 6 MV and 15 MV, was ~4% for the no-mask case and condition #1. On the other hand, the difference between the no-mask case and condition #2 was approximately 23% for 6 MV and 22% for 15 MV.

Experimentally, the phantom was irradiated by a head and neck treatment plan using direct fields with conventional fractionation of 200 cGy. The treatment plan was calculated with the Analytical Anisotropic Algorithm (AAA), in order to perform a comparison using Monte Carlo simulations in MCNPX. Using Equation (1), the absorbed dose of the 12 positions of the radiochromic films in the anthropomorphic phantom was calculated, finding that in positions 1–6, the absorbed dose does not present a significant bolus effect. This is due to the fact that the treatment plan chosen for this study did not have irradiation fields that provided significant absorbed dose in the areas where these films were located. On the other hand, the films located in positions 7–12 received significant absorbed doses that allowed a detailed study of the absorbed dose. The results are recorded in Table 2 along with the respective errors associated with the polynomial fit.

In Table 2, the increase in absorbed dose was calculated, where M represents the cases with the mask and S, the cases without the mask, finding that in positions 8 and 11, the highest associated bolus effect was observed. Moreover, the position with the lowest bolus effect was position 9.

## 3. Discussion

In the case of 6 MV photons, the theoretical results of this investigation are consistent with other studies, which find that the bolus effect generated by the thermoplastic mask can increase the surface absorbed dose by between 9% and 36% [9] and between 8.7% and 31.5% [7]. These findings indicate that the results found here are consistent with what has been published so far.

Regarding the 15 MV photon spectrum, the literature presents more important differences in the repercussions of the bolus effect due to the use of the thermoplastic mask. While some authors consider that there is a small degree of dosimetric uncertainty, even suggesting the possibility of ignoring the bolus effect [4,10], other authors, on the contrary, find increases in absorbed dose ranging from 6% to 22% [9], aligning more with the results found in this study.

Published research on this topic warns that the bolus effect generated by the thermoplastic mask is more pronounced for lower energies. Our results show a slight difference in the increase in surface absorbed dose for 6 MV energy compared to 15 MV energy. However, we found a notable result: the PDDs with 15 MeV energy experienced a greater displacement. This is evidenced by comparing the displacements of the maximum absorbed dose points in both cases: for 15 MV, it is displaced by 6 mm, and for 6 MV, by 3 mm.

Experimentally, for 6 MV photons, the bolus effect shows an increase of ~19.2%, while Monte Carlo calculations show an increase of less than 7.5%. This difference is associated with the fact that the radiochromic films used for the measurements were placed in different positions of the phantom: experimentally, in the anterior part of the cheek and with the simulations, in the upper part of the head. In contrast, the theoretical and experimental results for the chin position are consistent in terms of the increase in absorbed dose: ~42% in both cases. However, this anatomical position cannot be considered the one with the highest bolus effect, as was theoretically expected. The results are consistent with the chosen treatment plan, where the highest absorbed dose corresponded to the left lateral position, and the position of the highest experimental bolus effect was very close to the target to be irradiated. Beyond this, it becomes evident that the stretching conditions of the mask in the chin position make it one of the anatomical positions with the highest bolus effect.

## 4. Materials and Methods

This work comprises a theoretical study and an experimental corroboration as follows.

### 4.1. Monte Carlo Calculations

Simulations were performed using the MCNPX version 2.6 code [13]. The source term used corresponds to the spectra shown in Figure 3.

The geometric characteristics of the mask are shown in Figure 1b. This corresponds to a volume that varies according to Table 3. Structurally, it is filled with ICRU material [15]. This filling is performed because the mask, when in contact with the patient, allows part of the skin to penetrate the holes during molding, especially with the patient’s face. This recreates a realistic geometry.

Three masks used by patients in head and neck cancer treatments were considered. Two positions were identified where the highest and lowest bolus effects were expected: the chin and top of the head, respectively. Thickness measurements, the number of holes per square centimeter, hole radius, and percentage of hole occupation in the mask’s surface area were obtained in these positions (refer to Table 3), allowing for a faithful reproduction of the mask geometry in the simulation.

Initially, the radiation beam tuning process was carried out, which consisted of calculating the Tissue Phantom Ratio (TPR) at the depths of 20 and 10 cm (TPR 20/10) for the theoretical PDD, which was compared with the experimental PDD. Values of 0.564 and 0.558 were obtained for 6 MV, and 0.631 and 0.637 for 15 MV, respectively. Figure 4 shows the tunings for the two radiation beams.

### 4.2. Elemental Composition and Density of the Thermoplastic Mask

Klarity^®^ U-Frame-type masks were used [16], whose elemental composition was identified using Scanning Electron Microscopy (SEM) CARL ZEISS Model RA-ZEI-001, Fourier Transform Infrared Spectroscopy (FTIR), and X-ray Fluorescence (XRF) techniques. The analysis results show that the material of the mask corresponds to the base polymer C_10_H_16_O_4_, which effectively corresponds to Polycaprolactone (PCL). This material presents important characteristics with a glass transition temperature of about 60 degrees Celsius and low toxicity, which makes it ideal for biomedical applications with immobilization objectives, allowing adaptation to irregular morphologies. This conclusion was based on several factors: the SEM technique found that the carbon/oxygen ratio was 5:2, and the FTIR technique identified several polymers with similar spectra, discarding those that did not meet this ratio. Furthermore, the XRF study evidenced the presence of sulfur in the material, suggesting its possible inclusion.

After identifying the presence of sulfur, it was included in the tuning to corroborate possible changes in the PDD, using the weight fractions shown in Table 4. Figure 5 shows the tuning with and without sulfur, demonstrating that the presence of sulfur does not significantly alter the PDD.

To calculate the mask density, the volume occupied in water was measured, obtaining a value of 105.4 cm^3^, with a weight of 118.7 g, resulting in a density of 1.12 g·cm^−3^. Comparing with the Computed Tomography (CT) scan performed on the mask, it was found to have −315 Hounsfield Units (HU), indicating a density lower than 1 g/cm^3^. This, although it may seem a contradiction, has an explanation: the mask changes its density when molded over the patient. This was observed when it sank in water after its first use. This finding is supported by studies indicating that the mask’s density can vary between 0.9 and 2 g·cm^−3^ [5].

### 4.3. Geometry Construction

The calculation geometry was constructed as a phantom of ICRU material with the dimensions 50∙50∙50 cm^3^ composed of 10.1% H, 11.1% C, 2.6% N, and 76.2% O [15]. Figure 6a schematically shows the full geometry used. On the central axis of the phantom, calculation voxels of 0.9 cm^3^ are distributed on the surface, increasing in volume with increasing depth. At an SSD of 100 cm, the phantom is impacted with the photon spectra of 6 and 15 MV shown in Figure 3. The mask simulation on the phantom is shown in Figure 6b, considering the thicknesses and filling diameters of the different holes according to Table 3.

### 4.4. Blackening Scale Created for Reading Radiochromic Films

A scale was constructed for reading by irradiating the 12 radiochromic films one by one, located at a depth of 5 cm within a solid water phantom, with absorbed doses ranging from 20 to 240 cGy. These data are shown in Table 5, where the absorbed dose (D), measured with radiochromic films, and the Blackening level (B) after irradiation are displayed. Subsequently, using the Mephysto software, version 3.4 [17], a model was created to transform the percentage of Blackening of the films into absorbed dose. The correlation of the polynomial fit was R^2^ = 0.99, with an average uncertainty of 1.03%. Figure 7 shows the resulting fit for reading the radiochromic films.

The particular equation found corresponds to
B = 2^−6^∙D^3^ − 1.6^−3^∙D^2^ + 0.4377∙D + 55.146(1)
where B is the percentage of Blackening (%) and D is the absorbed dose (cGy).

### 4.5. Experimental Setup

Twelve radiochromic films were placed at strategic points on an anthropomorphic head phantom, as shown in Figure 8. The phantom system, mask, and radiochromic films were irradiated using a 6 MV photon treatment plan on a Clinac IX in two stages: with a mask and without a mask.

## 5. Conclusions

It was observed, both experimentally and theoretically, that the chin is one of the anatomical positions where a significant bolus effect can be expected for this particular treatment. On the other hand, theoretically, the bolus effect on the top of the head is smaller and can be ignored.

Theoretically, it was found that for 6 MV, the surface absorbed dose increases by between 4.1% and 23.2%, while the point of maximum absorbed dose shifts towards the surface by 0.3 ± 0.15 cm.

For 15 MV, the surface absorbed dose increases by between 4.26% and 22.56%, while the point of maximum absorbed dose shifts towards the surface by 0.6 ± 0.15 cm. These results confirm that for both 6 MV and 15 MV, the bolus effect can significantly increase the surface absorbed dose and should be considered in dosimetric planning systems.

## Figures and Tables

**Figure 1 ijms-25-09133-f001:**
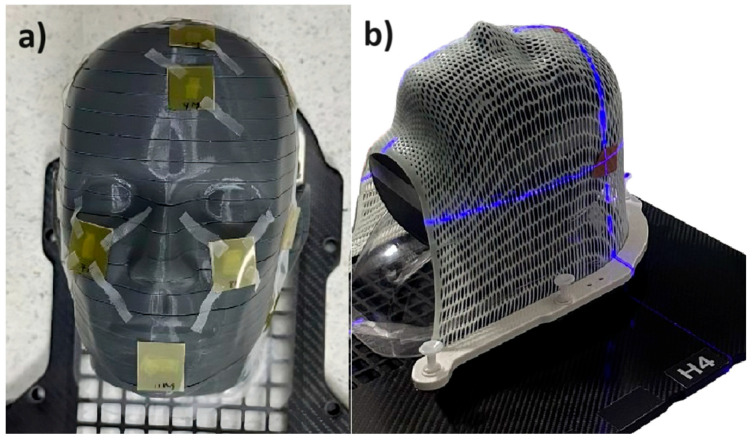
Panel (**a**) shows the phantom without a mask, with the radiochromic films located at the points of interest. Panel (**b**) shows the phantom with a mask.

**Figure 2 ijms-25-09133-f002:**
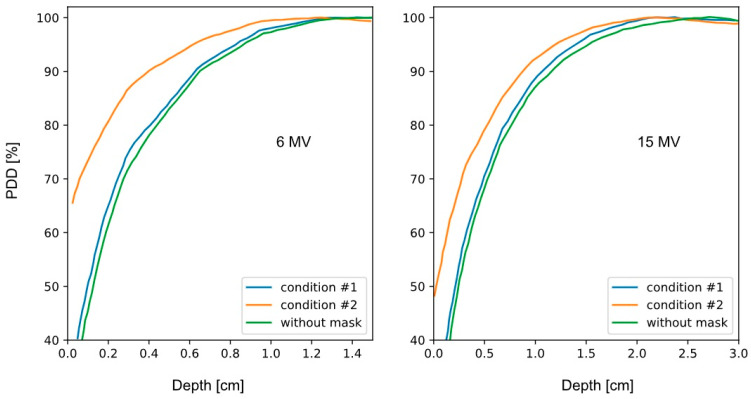
Monte Carlo simulations in MCNPX for studying the bolus effect with 6 and 15 MV energies.

**Figure 3 ijms-25-09133-f003:**
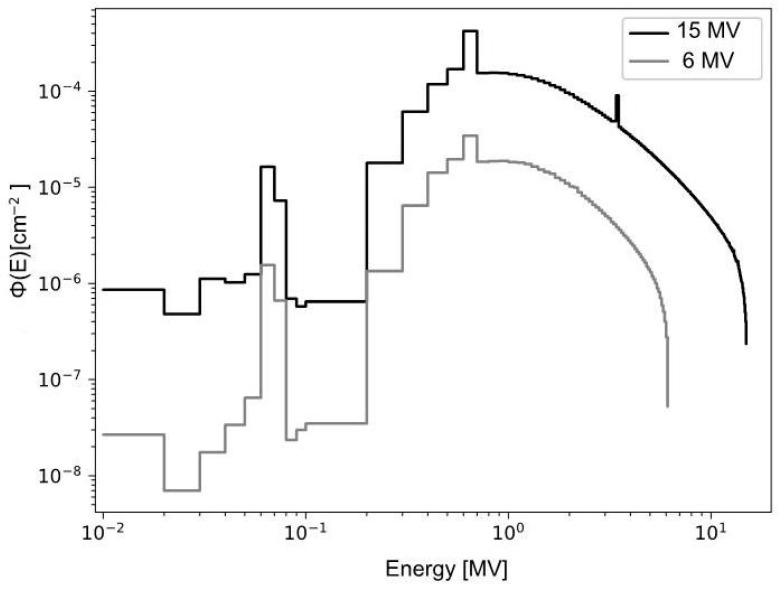
Spectra of 6 and 15 MV from Varian IX linear accelerator [14].

**Figure 4 ijms-25-09133-f004:**
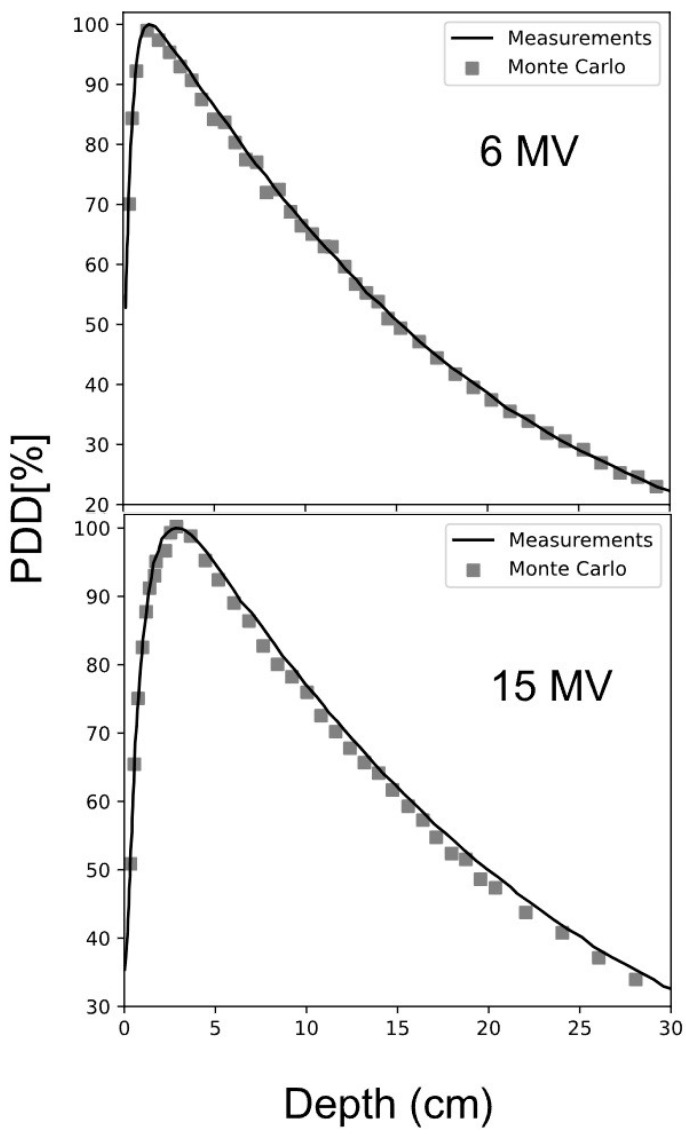
Tunings of 6 and 15 MV radiation beams.

**Figure 5 ijms-25-09133-f005:**
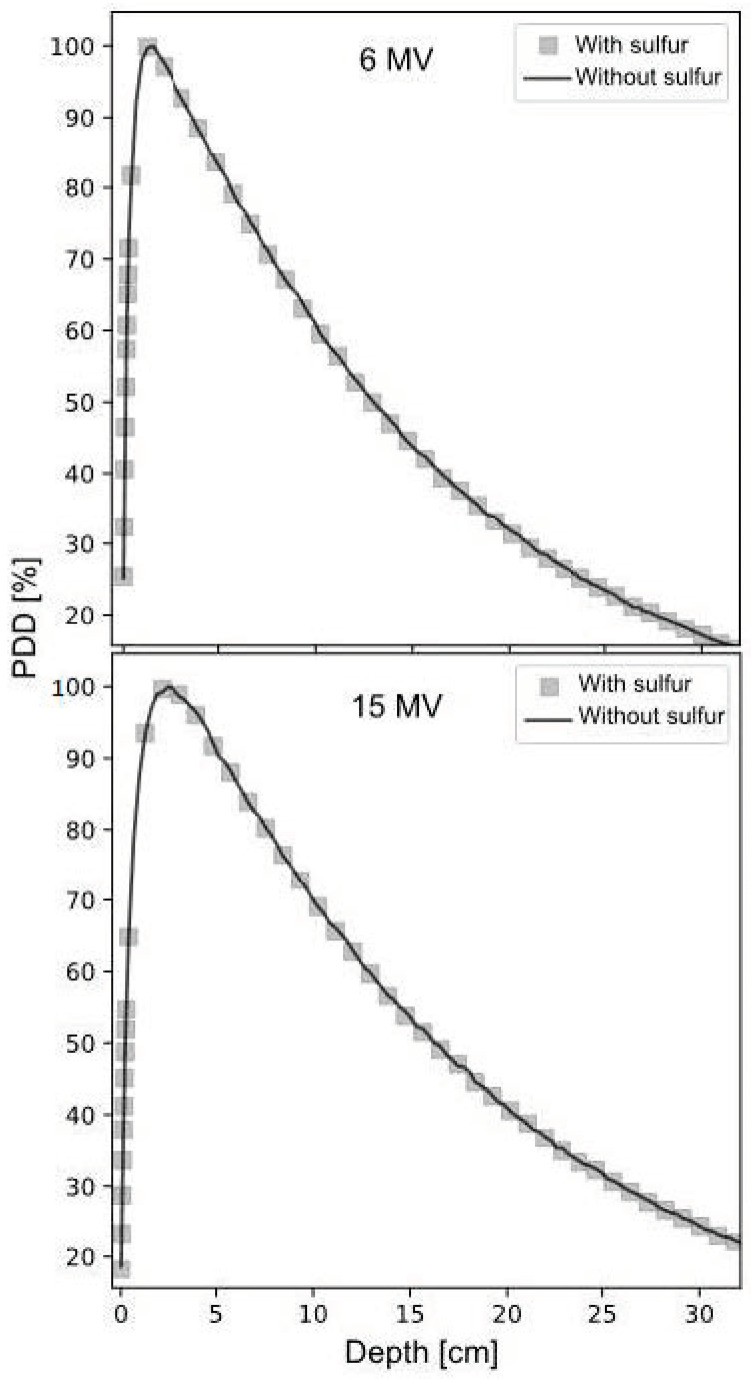
PDD calculation with and without S for 6 and 15 MeV energies.

**Figure 6 ijms-25-09133-f006:**
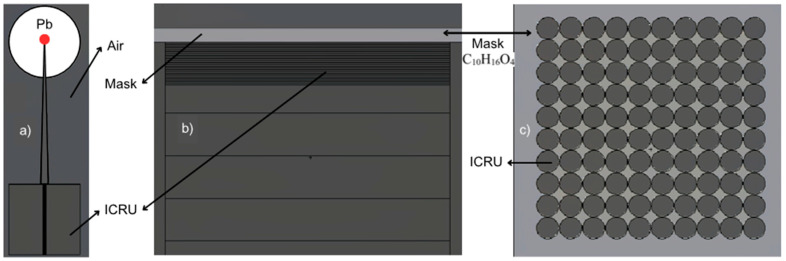
Two-dimensional diagrams of the geometry used in the simulations. (**a**) General diagram in the XZ plane where the isotropic source in red can be observed, shielded by a lead sphere with a pyramidal opening reflecting a 10 × 10 cm^2^ field on the phantom surface at a source to surface distance (SSD) of 100 cm. This geometry is repeated for calculations with and without a mask, conditions #1 and #2 described in Table 3. (**b**) XY plane of the voxel distribution in the phantom, corresponding to fine voxelization on the surface and coarser voxelization after the first 3 mm. (**c**) Hole distribution for condition #1 detailed in Table 3.

**Figure 7 ijms-25-09133-f007:**
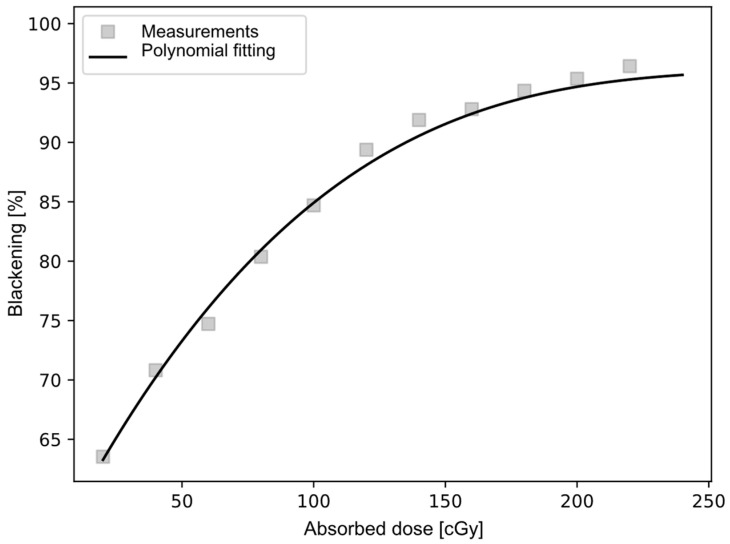
Polynomial fit used for reading the radiochromic films.

**Figure 8 ijms-25-09133-f008:**
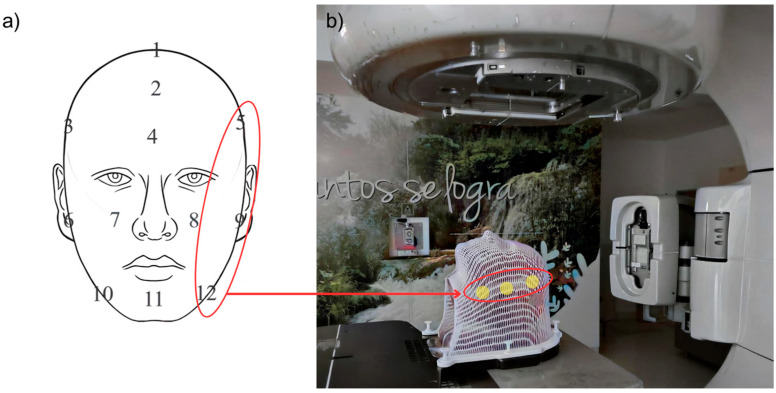
Experimental setup. (**a**) Positions of the 12 radiochromic films on the anthropomorphic phantom. (**b**) Experimental setup indicating reference radiochromic films 5, 9, and 12 for reading.

**Table 1 ijms-25-09133-t001:** Representative results of the main characteristics studied in the simulations reproducing Figure 1. Condition #1 corresponds to thicknesses of 1 ± 0.1 mm (Head) and 2.6 mm ± 0.1 mm (Chin).

Energy (MV)	Condition	Surface Dose (%)	Maximum Dose Position (±0.15 cm)
6	Without Mask	54.41	1.55
6	#1	58.5	1.55
6	#2	77.6	1.25
15	Without Mask	38.24	2.75
15	#1	43.5	2.75
15	#2	61.8	2.15

**Table 2 ijms-25-09133-t002:** Experimental values of Blackening (%) and their respective absorbed dose values, found through Equation (1), along with the error associated with the polynomial fit. Finally, the percentage increase compared to the expected surface dose without a mask is presented for the films located in positions 7–12.

#	Blackening (%)	Absorbed Dose (cGy)	Increase (%)
M-7	87.9	118.7 ± 6.1	33.6
S-7	82.8	88.8 ± 5.7	
M-8	88.1	120.1 ± 6.1	47.2
S-8	81.3	81.6 ± 5.8	
M-9	80.1	76.2 ± 5.9	19.2
S-9	77.1	63.9 ± 6.2	
M-10	88	119.4 ± 6.1	30.1
S-10	83.4	91.8 ± 5.7	
M-11	90.3	137.7 ± 6.7	42.54
S-11	84.3	96.6 ± 5.7	
M-12	91.7	146 ± 7.24	34.1
S-12	86.42	108.9 ± 5.9	

**Table 3 ijms-25-09133-t003:** Parameters used for constructing the thermoplastic mask geometry in the simulation, detailing two conditions for the highest and lowest expected bolus effects, chin and top of the head, respectively.

Condition	#1 (Head)	#2 (Chin)
Thickness (mm)	1 ± 0.1	2.6 ± 0.1
Hole occupation (%)	74.8 ± 1	21.8 ± 2.5
Holes/cm^2^	1 ± 0.15	9 ± 1
Radius Hole (mm)	4.88 ± 0.2	0.87 ± 0.2

**Table 4 ijms-25-09133-t004:** Weight composition of the materials comprising the mask.

Element	C	H	O	S
Without sulfur	0.5996	0.0805	0.3199	0
With sulfur	0.5168	0.0693	0.2756	0.1383

**Table 5 ijms-25-09133-t005:** Absorbed dose and Blackening for each of the 12 radiochromic films.

#	1	2	3	4	5	6	7	8	9	10	11	12
D (cGy)	20	40	60	80	100	120	140	160	180	200	220	240
Blackening (%)	63.5	70.8	74.7	80.3	84.7	89.3	91.8	92.8	94.3	95.3	96.4	99.6

## Data Availability

The datasets and materials used and/or analyzed during the current study are available from the corresponding author upon reasonable request.

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
