# Peer review of "Bolus Effect Caused by Use of Thermoplastic Masks in Head and Neck Radiotherapy Treatments"

_ijms, 2024, doi:10.3390/ijms25169133_

Round 1

Reviewer 1 Report

Comments and Suggestions for Authors

In this study, the material composition of thermoplastic masks was obtained through the application of various material characterization techniques. The bolus effect of thermoplastic masks in head and neck tumor radiotherapy was studied by Monte Carlo calculation and radiochromic radiation lms measurement, which has certain guiding significance for the formulation of head and neck tumor radiotherapy plan. The main problems in this article are as follows:

1) When the abbreviation appears for the first time, it needs to be spelled in full, such as TPR 20/10, PDD, etc.;

2) Pay attention to the standardization of spelling, such as +- in Table 1, need to be changed to ±;

3) Pay attention to the uniform standard of units. In this paper, some X-ray energy units are MV and some are MeV;

4) It is better to give pictures of thermoplastic masks before Table 1;

5) Please give details of the radiotherapy plan used when the radiochromic Radiation measurement is performed, and suggest changing the radiotherapy plan used or using multiple radiotherapy plans to cover more areas;

6) The main differences between head and neck radiotherapy and body radiotherapy may be caused by body surface masks. Because head and neck masks are more variable, the relationship between radiation and thermoplastic masks during radiotherapy is more complicated. It is suggested to consider the effect of thermoplastic masks on the body surface from the perspective of radiation and thermoplastic masks.

Reviewer 2 Report

Comments and Suggestions for Authors

The authors should highlight the advantages and drawbacks of CPL, as main constituent material in the thermoplastic mask. 

Moreover, additional studies could explore different mask materials and configurations, as well as the impact on deeper tissue doses.

Finally, incorporating patient-specific adjustments in the simulation models could enhance the accuracy of dose predictions.

Comments on the Quality of English Language

Minor editing of English language required. 

Round 2

Reviewer 1 Report

Comments and Suggestions for Authors

The manuscript has been sufficiently improved to warrant publication.